# Building Façade Color Distribution, Color Harmony and Diversity in Relation to Street Functions: Using Street View Images and Deep Learning

**Yujia Zhai \*, Ruoyu Gong , Junzi Huo and Binbin Fan**

College of Architecture and Urban Planning, Tongji University, Shanghai 200092, China
* Correspondence: zhai@tongji.edu.cn

**Abstract:** Building façade colors play an important role in influencing urban imageability, attraction and citizens' experience. However, the relations between street functions and the building façade color distribution, color harmony and color diversity have not been thoroughly examined. We obtained the dominant colors of building façades in Changning District, Shanghai, utilizing Baidu street view images, image semantic segmentation technology and the K-means algorithm. The variations in building façades' dominant colors, color harmony and diversity across different types of functional streets were examined through logistic regression and ANOVA analyses. The results indicate that, compared to industrial streets, red hues are more common in science education streets, residential streets and mixed functional streets. Business streets are more likely to have hues of green, red and red–purple. Residential streets' saturation is overall higher than that of industrial streets. In business streets, the medium–high value occurs less frequently than other streets. Moreover, we found that the street building façade colors in industrial streets were more harmonious and less diversified than that in other functional streets. This study has implications for urban color planning practices. Color harmony and color diversity should be well considered in future planning. The role of street functions should also be addressed in building façade color planning, to improve existing planning frameworks as well as related strategies.

**Keywords:** building façade color; street function; color harmony; color diversity; deep learning

## 1. Introduction

Building façade colors are the most critical colors in the urban space, playing an important role in influencing urban imageability, attraction, certain economic and practical issues [1] and citizens' experiences, as well as emotion and temperature perceptions [2]. Nowadays, measuring the criteria of color objectively and quantitatively is becoming more important in urban color planning [3]. Color planning comprises many parts; in the urban color landscape, it usually includes the colors of the natural landscape, building façades, roofs, advertising signs, public facilities, transportation tools and pavements [4]. The building façade accounts for the largest visible proportion of the building and can significantly impact the building's overall color appearance [5]. People's experience of the urban environment is shaped by its physical colors [6]. Color changes the citizens' understanding of the urban space and can change the emotional attachment between the citizens and the city [4]. Color can also enhance the continuity of the street and highlights the outline of the street.

Building façade colors are related to many variables, among which functional attributes are critical [7]; however, few studies have examined the relationship between building façades and street functions [8]. Meanwhile, building façade color analyses at the level of the street are also lacking [3]. As the basic unit of the city, streets are the primary sites for public activities and are important to urban environment perception. A failure to address functional issues and analyses at the street level may decrease the depth and

accuracy of building façade research, which may limit the application of related findings in urban color planning practices. Especially in China, urban color planning practices have difficulties in meeting the practical requirements of urban development [9]. Therefore, we need to address the relationship between building façade color analyses and street functions, in order to better shed light on related practical planning. Apart from the basic attributes of the building façade color, color harmony and diversity are also very important [10]. Color itself has a variety of attributes and planning principles. In most experimental studies, the most important of these properties are hue, saturation and brightness [11]. Others color attributes, such as harmony, diversity and pleasantness, are also related to street functions [11–13]. In modern architecture color applications, three principles are proposed by comparing three color composition systems. As for architecture colors, less hue, low saturation and high brightness, such as light greyish colors, are preferred [14].

Meanwhile, objective tools such as big data and machine learning in building façade color analyses have been applied in urban studies. Compared to traditional urban color study methods [3], these new tools can handle data rapidly and expand the scope of the research. New technologies in color studies involve street view image crawling, image semantic segmentation and color clustering [15–18]. Large-scale color extraction and analysis methods of building façades have become an important basis for urban color planning and data-driven color management.

The present study has three main contributions. First, street functions are addressed in examining the building façade color. Second, we also explore color harmony and color diversity among the building façade's dominant colors from the perspective of street functions. Third, we propose an approach integrating advanced technologies, such as crawling street view images, image semantic segmentation and the K-means algorithm, which are concretely applied to explore the street building façade color distribution based on street functions in Changning District, Shanghai.

## 2. Literature Review

### 2.1. Street Function and Urban Color Distribution

Existing studies indicate that many factors are related to the urban color distribution, such as building types [7], the urban history context [19,20], the spatial configuration [9], aesthetic preferences [21], ethical conceptions [14], citizens' education levels [15,22] and emotions [23], etc. Researchers propose several frameworks related to urban color. Hong and Ji argued that a city's main color was influenced by the partitioning of city functions [4]. Gou argued that color planning is related to geography, tradition, technology, layout, function, synthesis, culture, values, economy, psychology, flexibility, sustainability, aesthetics and the increased urban density (vertically and horizontally) [24]. Reyhaneh indicated that urban politics, the economy and culture can impact the city's colors, so we should combine the roles, functions and scales of the city with the environment's colors [25]. Wang, Zhang et al. identified six factors affecting color, which are the color combination, building volume, style, material, function and geographical location [26]. The above studies indicate that the building function is an important factor related to the city color. Building functions generally include six categories, namely housing, public services, commercial services, transport facilities, landscaping and public utilities [19].

Numbers of different types of buildings have increased with economic development and population growth in Shanghai. For instance, the growth trend of educational buildings is very stable, which highlights the need to consider the building façade color in relation to its functions. Blue and brick red are mainly used in high-rise residential buildings, while white is more often used in residential and public buildings [27]. The colors of residential buildings are mostly warm colors, with high brightness and low saturation [26]. At present, the color distribution is associated with a full color hue, low saturation and medium–high brightness in Shanghai [22]. In terms of hue, the historical conservation area is dominated by warm colors such as red–yellow, while the commercial areas have more cool colors, such as blue. In terms of saturation, the overall saturation is low. The developed urban

sub-centers have higher color saturation than their surrounding areas. The brightness of urban construction areas is lower than that of rural areas, with obvious differences between urban and rural areas [28]. Many studies addressing urban colors have been conducted in different areas in Shanghai. However, Changning District has not been addressed in previous studies.

### 2.2. Building Façade Color Measurements: Harmony and Diversity

Building façade color harmony and diversity can significantly influence the perceptions and preferences of the urban environment. Various methods have been proposed to measure color harmony and diversity. Regarding harmony, when two or more colors seen in neighboring areas produce a pleasing effect, they are said to produce color harmony. Many evaluation models addressing color harmony between two colors have been proposed. For example, Ou and Luo proposed a color harmony evaluation formula based on the evaluation of 1431 color pairs by 17 people [29]; other models proposed by Nayatani and Sakai [30] and Szabó et al. [31] are also widely used. For color harmony among three colors, the Szabó model was created based on the assessments of 14,280 groups of three-color combinations, and Ou et al.'s method relies on 6545 clusters of three-color combinations [32]. Advanced models also apply artificial intelligence to measure color harmony, such as Gated CNN and the Siamese network [33]. Existing research identifies a few common principles; for instance, colors with equal hue, equal chroma and unequal lightness values tend to appear harmonious [34]. The higher the lightness value of one color in a color pair, the more likely they are to appear harmonious. Color diversity could be measured by the fractal dimension algorithm using the box-counting method [35]. Fractal dimension and multifractal dimension are used as a measure of color diversity [36].

### 2.3. Advanced Technology in Urban Color Studies

The development of new technology can significantly advance urban color research and planning practices [24], such as street view images, semantic segmentation and color clustering, etc. New technologies can advance color extraction, color management and analysis in urban color analyses and planning. Street view images' color extraction involves two steps, namely obtaining the color and correcting the color [37]. Traditional methods such as chromatography, colorimetry and instrument color measurement have many disadvantages, such as that the sample sizes are small, the time costs of color selection are high, and the interference of subjective judgment is large [38]. Nowadays, new techniques such as deep learning are increasingly applied to color extraction research [19]. For example, we can use web crawlers to obtain street view images and segment images using semantic segmentation technology [18]. In this context, the PSPNet Python model is more common as a semantic segmentation tool. Related color identification and extraction methods include urban color clustering extraction methods and algorithms, such as K-means [17], the cluster to standard color-chart [19] and the dominant color descriptor (DCD) [16], etc. Among them, the K-means is algorithm is widely used.

## 3. Materials and Methods

Our study included four main parts, as described in Figure 1. First, we obtained street view images of Changing District. Second, we used AWB methods to calibrate the images and extracted the building façades' dominant colors by K-means clustering, by which we preliminarily described the colors in three channels and determined the spatial distribution. Third, street functions were identified based on the frequency density of point of interest (POI) data. Finally, we explored the relationships between the building color distribution, color harmony, color diversity and street functions using a logistic regression model and one-way ANOVA.

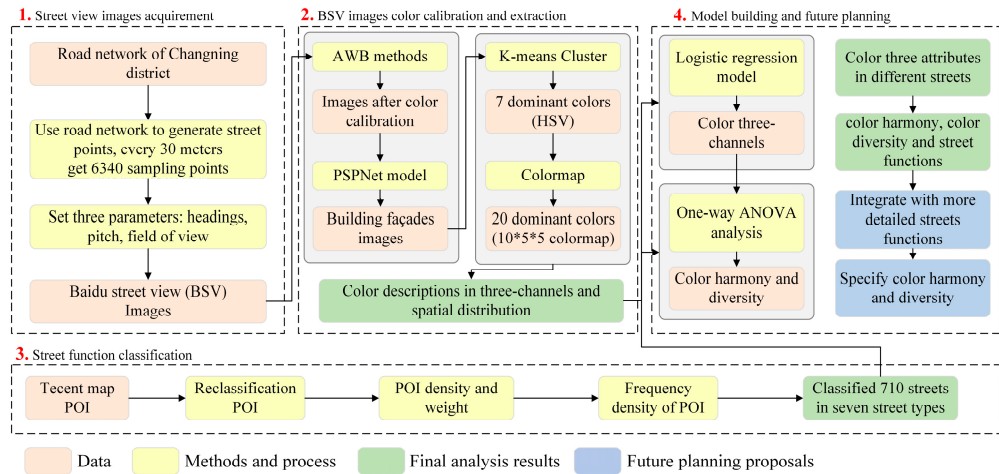

**Figure 1.** Flowchart of data processing and research method.

### 3.1. Study Area

This study was conducted in Changning District, Shanghai, China, as shown in Figure 2. Shanghai is the second-largest city in China, with an area of approximately 6340.5 km² and 24.9 million people. Changning District is located in the central district of Shanghai; it covers an area of 37.2 km² and had a population of 0.6 million people by the end of 2022 (Shanghai Municipal Statistics Bureau, 2022). Changning District is one of the central districts of Shanghai. The geographical conditions in Changning are representative; it has a longitudinal range from 121°26′01″ to 121°19′36″ E (WGS84 Universal Mercator Projection) and is characterized by a mild and humid climate, with an annual mean temperature of 19 °C. Suzhou River runs to the east and west of the area. Changning District's streets differ in color, types and features, and they could be a representative example of Shanghai's streets. Changning's streets are located across the inner ring, the middle and the outer ring of Shanghai. There are also historical blocks, as well as the Shanghai Hongqiao International Airport, in Changning District. The use of Changning District as a typical case for urban façade color description and analysis can guide future urban color planning.

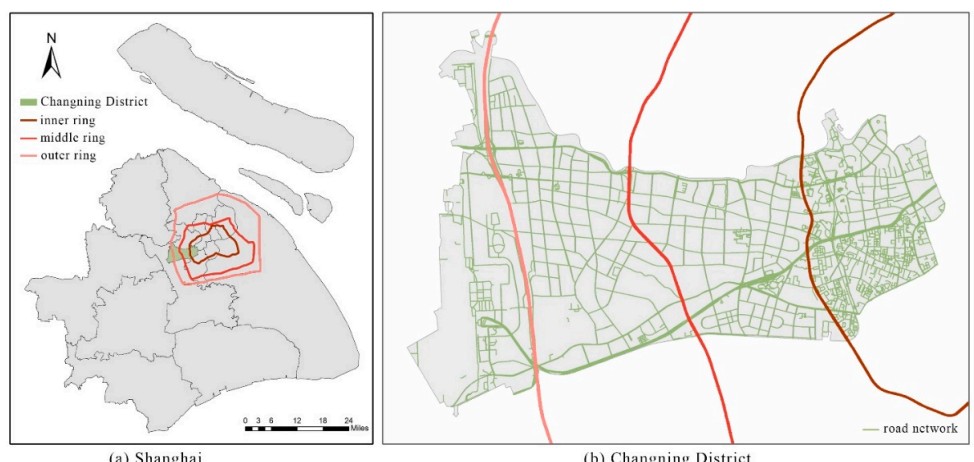

**Figure 2.** The location of Changning District in the city of Shanghai.

### 3.2. Street View Image Acquisition and Segmentation

We used semantic segmentation techniques to obtain buildings based on a large number of street view images and identified their main colors. Street view images of the same areas were usually taken at the same time sequentially, which could ensure the consistency of daylight, as well as simulating people's visual perception when walking along



streets [38]. We obtained Baidu street view (BSV) images utilizing the application program interface (API) from the Baidu Maps open platform (https://lbsyun.baidu.com/) [39] in September, 2022. The digital photographs were acquired from the BSV API with a resolution of 480 by 320 pixels. The process of crawling street view images included three steps; we simplified the route of the road network into a single line using ArcGIS [40]. We obtained the images at the distance of 30 m along the roads. In total, 7050 sample points on 752 roads were included, as seen in Figure 3a. Second, the WGS84 coordinates of the road network were converts into BD09 to ensure the location's accuracy. We captured the BSV images based on the latitude and longitude coordinates of the sampling points. All the image parameters were set as follows: headings = (0°, 90°, 180°, 270°), field of view = 90°, pitch = 0. These simulated the visual perception of pedestrians horizontally in four directions on the street [41], as shown in Figure 3b. A total of 26,151 images were crawled. Third, 800 images of the city's elevations and tunnels were excluded. Finally, we used 25,337 images (6340 sampling points) to identify building façade colors. As some streets had no buildings, 6276 sampling points were included in the color analysis.

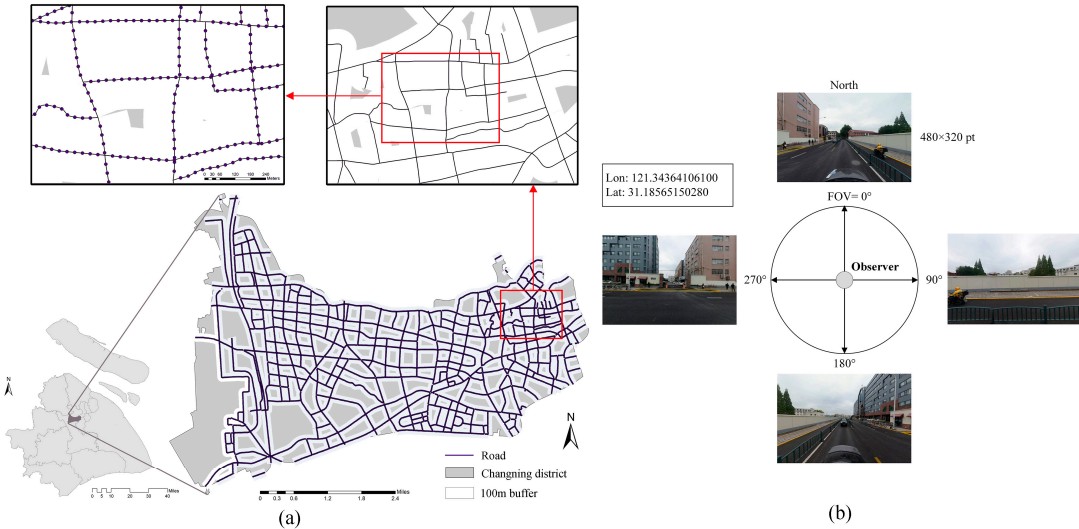

**Figure 3.** (**a**) Sampling points on the roads; (**b**) crawling images at the sampling points.

This study used the trained PSPNet recognition model based on the ADE20K dataset to semantically segment the Baidu street view images [42]. The model's accuracy (mIoU) rate was 85.4%; thus, it could ensure high accuracy in building façade identification. Semantic segmentation required classifying each pixel into a given set of categories [43]. Each pixel was classified into a category, such as building façades, trees, sky and sidewalks, as seen in Figure 4.

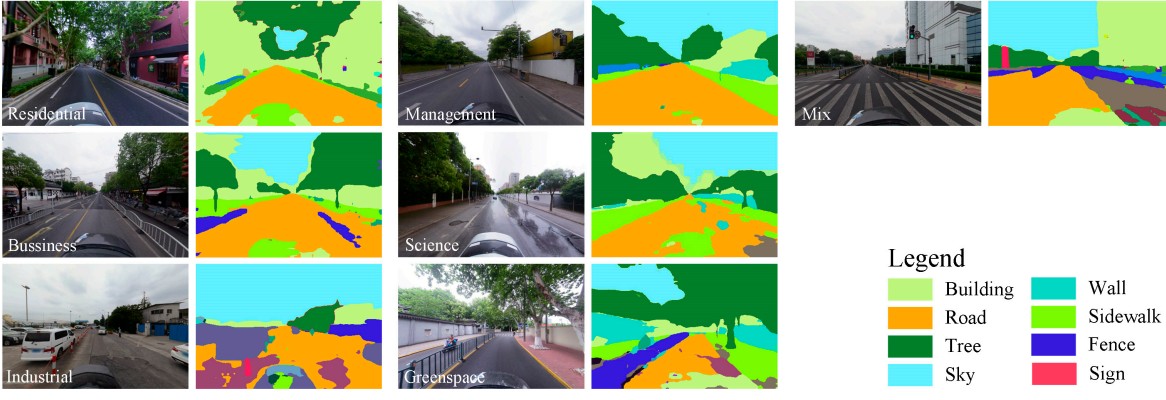

**Figure 4.** Segmented street view images based on PSPNet.

*3.3. Building Façade Color Calibration and Extraction*

### 3.3.1. Color Calibration

Sunlight, illumination and reflectance can significantly influence the colors captured in street view images. Therefore, color calibration was necessary. The method of automatic white balancing (AWB) can estimate illuminance and calibrate the image color automatically; thus, it is widely applied [44]. This method utilizes various algorithms based on different situations, such as the white patch, gray world, iterative white balancing, illuminant voting and color by correlation algorithms [45].

In this study, we used an algorithm combining the gray world and the white patch methods in AWB, in order to calibrate the building colors in the street view images [46]. The method of white patch assumes that the brightest color in an image should be close to white, and the color of the entire image should be corrected accordingly. The method of gray world assumes that the mean of the red, green and blue channels in a given scene should be roughly equal, to construct a grey color [47]. Specifically, Formula (1) below was used in our calibration. It denotes a full-color image of size $n \times n$ as $RGB_{sensor}(x, y)$, where $x$ and $y$ are the pixel positions, and $\mu$ and $v$ are parameters to be calculated, which can solved by a matrix [46]. We first converted the collected images from RGB to the CIElab color space, to better measure the different perceptions of the colors. We corrected each pixel in the image according to $\mu$ and $v$, as seen in Figure 5.

$$\begin{bmatrix} \sum_{x=1}^{n} \sum_{y=1}^{n} R_{sensor}^2(x,y) & \sum_{x=1}^{n} \sum_{y=1}^{n} R_{sensor}(x,y) \\ max_{x,y} R_{sensor}^2(x,y) & max_{x,y} R_{sensor}(x,y) \end{bmatrix} \begin{bmatrix} \mu \\ v \end{bmatrix} = \begin{bmatrix} n^2 G_{avg} \\ G_{max} \end{bmatrix} \tag{1}$$

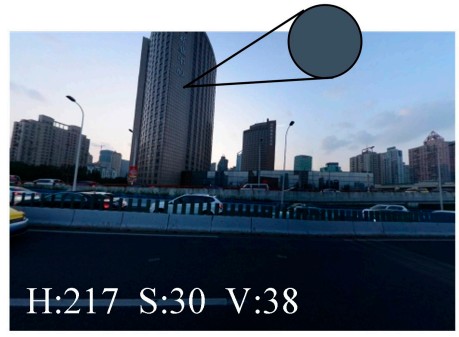 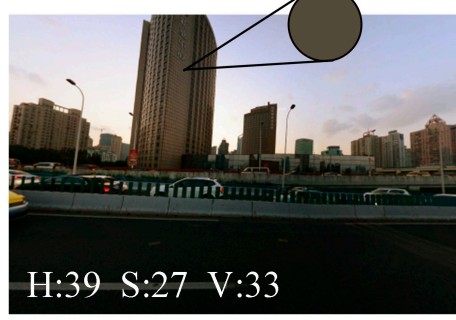

H:217 S:30 V:38      H:39 S:27 V:33

(a) original street view image      (b) after color calibration

**Figure 5.** Comparison of the original street view image and that after color calibration.

### 3.3.2. Extracting Building Façades' Dominant Colors

We excluded the colors with values of less than 50 in all three RGB channels, to diminish the possible influence of shadows, and then we used the mini-batch K-means algorithm in Python to extract the seven dominant building façade colors. The K-means algorithm has been widely applied to extract the main colors of urban landscape features and buildings [17]. Its main color extraction steps are as follows: (1) randomly select k colors to form k categories; (2) pixels in the image are assigned to the category with the shortest color distance; (3) calculate the Euclidean distances of the pixel colors under each category, to constitute a new category; (4) if the new category is consistent with the original one, end the calculation; if not, repeat steps 2 and 3 until they are consistent; (5) finally, each pixel in the image is assigned to the category with the shortest color distance to complete the dominant color extraction, as shown in Figure 6.

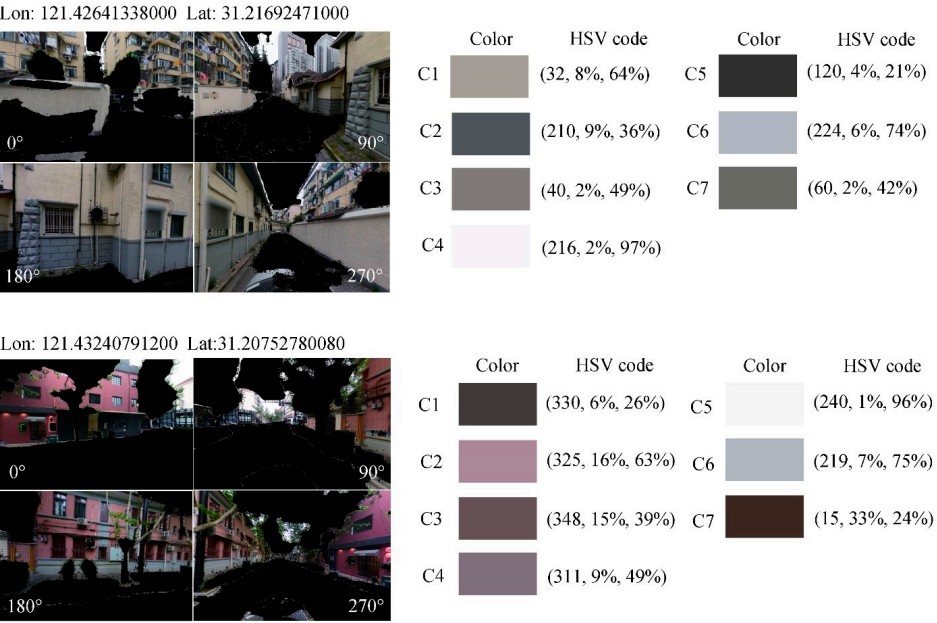

**Figure 6.** Seven dominant colors of building façade in the sample street.

### 3.3.3. Measurements: HSV Index, Color Harmony and Color Diversity

We measured the urban color characteristics with three measurements, the HSV index, color harmony and color diversity. We calculated the harmony of dominant colors based on the CIELab color space, utilizing Ou and Luo's formula, as shown in Formula (2) [29], which is based on the color perceptions of 1333 pairs of colors obtained from 54 participants. Color diversity was measured by the fractal dimension following previous studies, as expressed in Formula (3) [35].

$$CH = -2.2 + 0.03L_{sum} + H_{\Delta c} + H_{\Delta L} + 1.1H_H \tag{2}$$

$$d_{box}(\varepsilon) = -\lim_{\varepsilon \to 0} \frac{\log N(\varepsilon)}{\log(\varepsilon)} \tag{3}$$

### *3.4. Classification of Street Functions*

We divided the street functions into seven categories: residential, business, industrial, management and public services, science education, greenspace and mixed functional streets. We classified the street types based on the density and weight of POI, as in the following Formula (4). Here, $F_i$ is the frequency density, $W_j$ is the sum of weights of class $j$ POI in the buffer and $d_j$ is the number of class $j$ POI.

$$F_i = W_i \times \frac{d_i}{\sum_{j=1}^{6} (W_j X d_j) \times 100\%} \tag{4}$$

Street classification included four main steps. (1) We created 100 m buffer zones along the street network—there were 23,250 POI in total within the buffer zones. (2) We summarized the POI numbers in the buffer zones along each street segment. (3) According to Ziyi Wang and Debin Ma's study, we determined the POI weights [48] and used Formula (4) to identify single functional streets and mixed functional streets. When the frequency density of POIs in the buffer was greater than or equal to 50%, the functional street was regarded as a single functional street. Then, the other streets were defined as mixed functional streets. For instance, if the proportion of a residential POI was 80% related to a street, the street was regarded as a residential street. If the proportions of residential and business POIs were 37% and 27%, we defined the street as a mixed street. (4) The accuracy of street function classification was verified by visual interpretation.

## 4. Results

### 4.1. Street Function Classification

We classified 710 streets into different functions, as seen in Figure 7. Among them, there were 151 residential streets (20.08%), 169 business streets (22.47%), 73 industrial streets (9.71%), 67 management and public services streets (8.91%), 20 science education streets (2.66%), 8 greenspace streets (1.06%) and 224 mixed functional streets (29.79%).

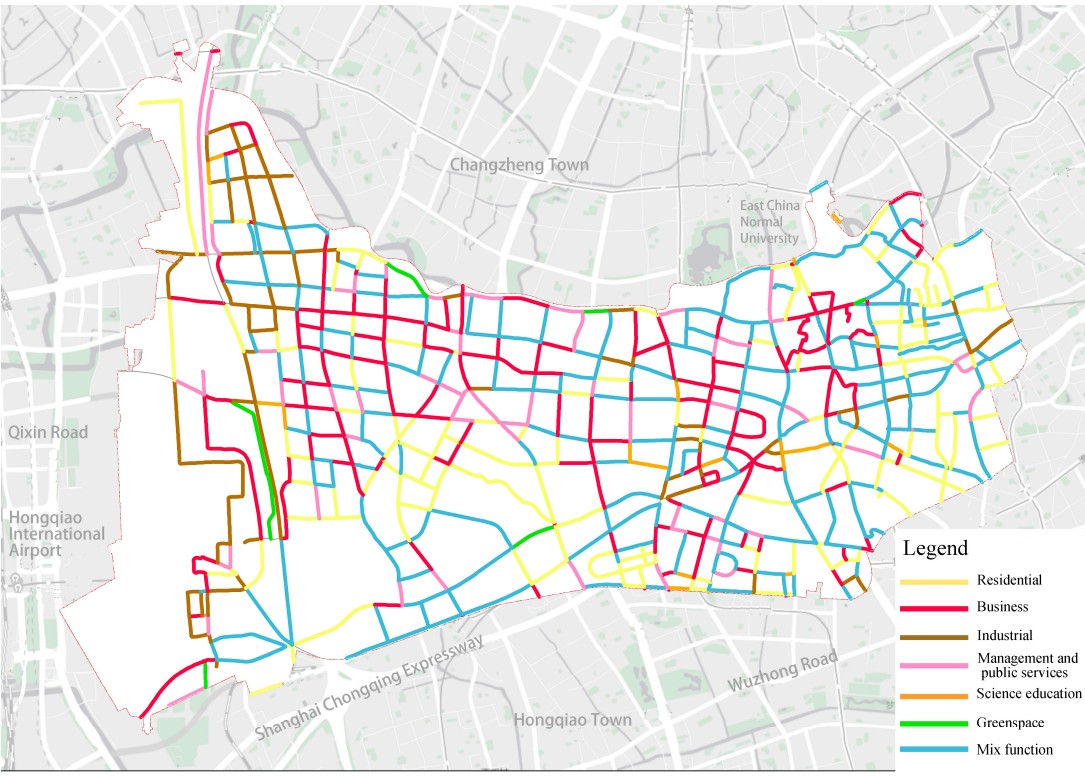

**Figure 7.** Street function classification in Changning District.

### 4.2. Building Façade Dominant Colors

4.2.1. Overall Color Distributions in Changning District

In total, 657 streets with 6276 sampling points were included in the analysis, and streets without street view images were excluded. For each sampling point, we identified the three most dominant colors. In HSV mode, the color hue is divided into ten categories based on the Chinese color system [49], resulting in five basic colors of red (R), yellow (Y), green (G), blue (B) and purple (P) and five intermediate colors of red–yellow (YR), yellow–green (GY), green–blue (BG), blue–purple (PB) and purple–red (RP). Saturation is divided into low saturation (L), medium–low saturation (ML), medium saturation (M), medium–high saturation (MH) and high saturation (H) [50]. The saturation's corresponding numerical values are 0–2, 2–10, 10–30, 30–60, 60–100. The value of colors is divided into 5 groups, including 0–26, 26–51, 51–89, 89–140, 140–255.

The hue includes mainly blue (B), red (R) and red–yellow (YR), which account for 18.7%, 18.1% and 16.7%, respectively. Yellow–green (GY) is the least prevalent, accounting for 3.08%. Saturation is mostly medium–low (ML), followed by Low (L), with ML accounting for 59.59%. Values are mostly classified as high (H), followed by medium (M), with high (H) accounting for 48.93% and medium (M) accounting for 26.17%. Colors with low saturation and high values occur the most on building façades, as shown in Figure 8.

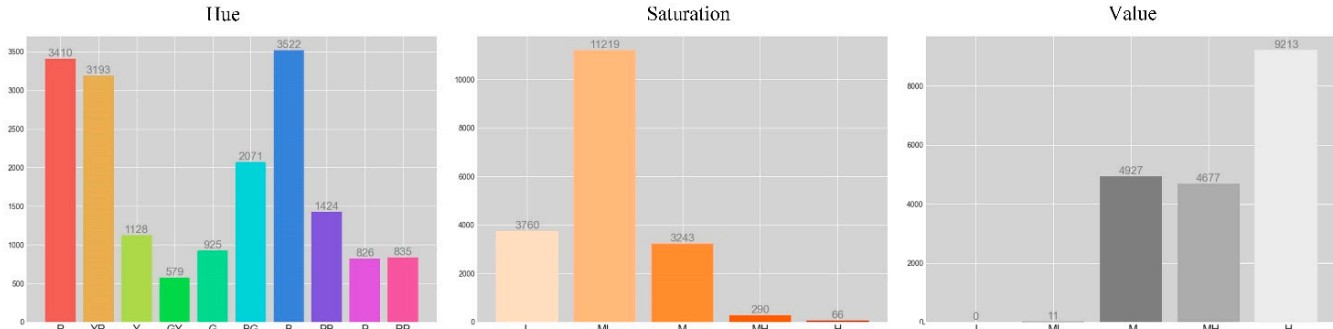

**Figure 8.** Hue, saturation and values of the building façade colors.

Hue is distributed differently across different values. Under a low value (V2), colors of red and red–yellow are the most frequent. When the value becomes higher, the blue color becomes more frequent; when the value increases to V4 and V5, the blue color becomes the most dominant, as seen in Figure 9a. Colors of red and yellow are generally associated with dark materials on building façades, such as bricks. The blue color is mainly related to glass windows. The most dominant colors are dark grey red (H1S1V2), light purple–blue (H7S1V5), dark grey yellow–green (H2S1V2), medium grey brown (H1S1V3), light grey blue (H6S1V5) and medium grey purple (H7S1V4), as shown in Figure 9b.

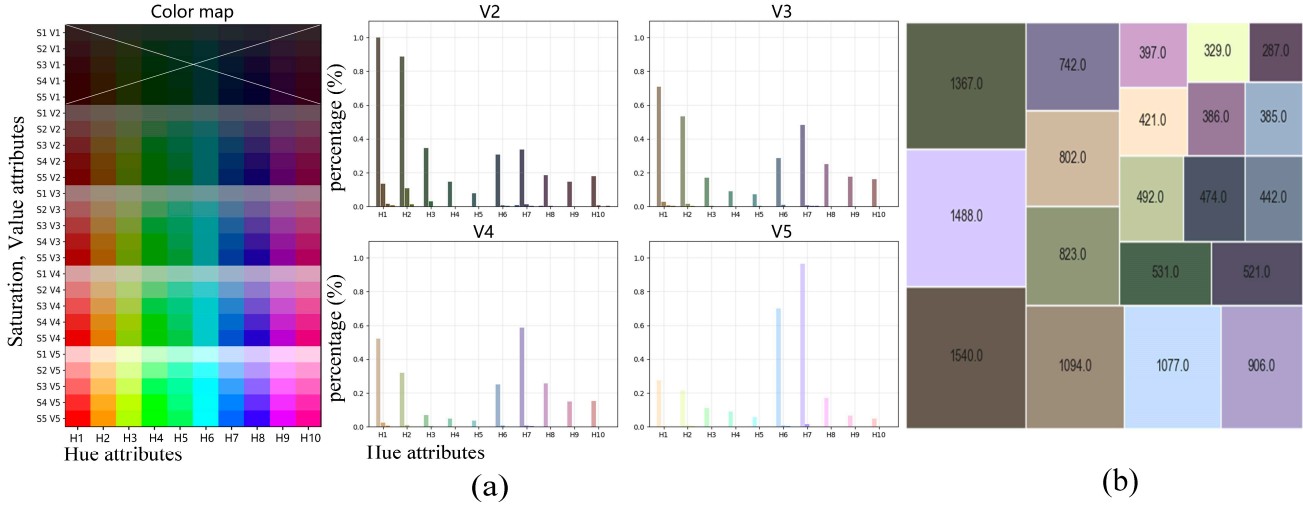

**Figure 9.** Building façade color distribution in Changning District. (**a**) Color map; (**b**) Tree map of dominant colors.

The dominant color varies between different blocks, as seen in Figure 10. We found some streets (1) with many residential buildings and supporting businesses, where yellow–brown colors appear the most. Hongqiao International Airport (2) and business and mixed functional streets (3) are dominated by blue colors, which might be related to the presence of glass materials. In the mixed residential and commercial streets (4), yellow–brown and light blue constitute almost half of all the colors. The outer ring's forest belt ecological greenway (5) is dominated by yellow–brown colors.

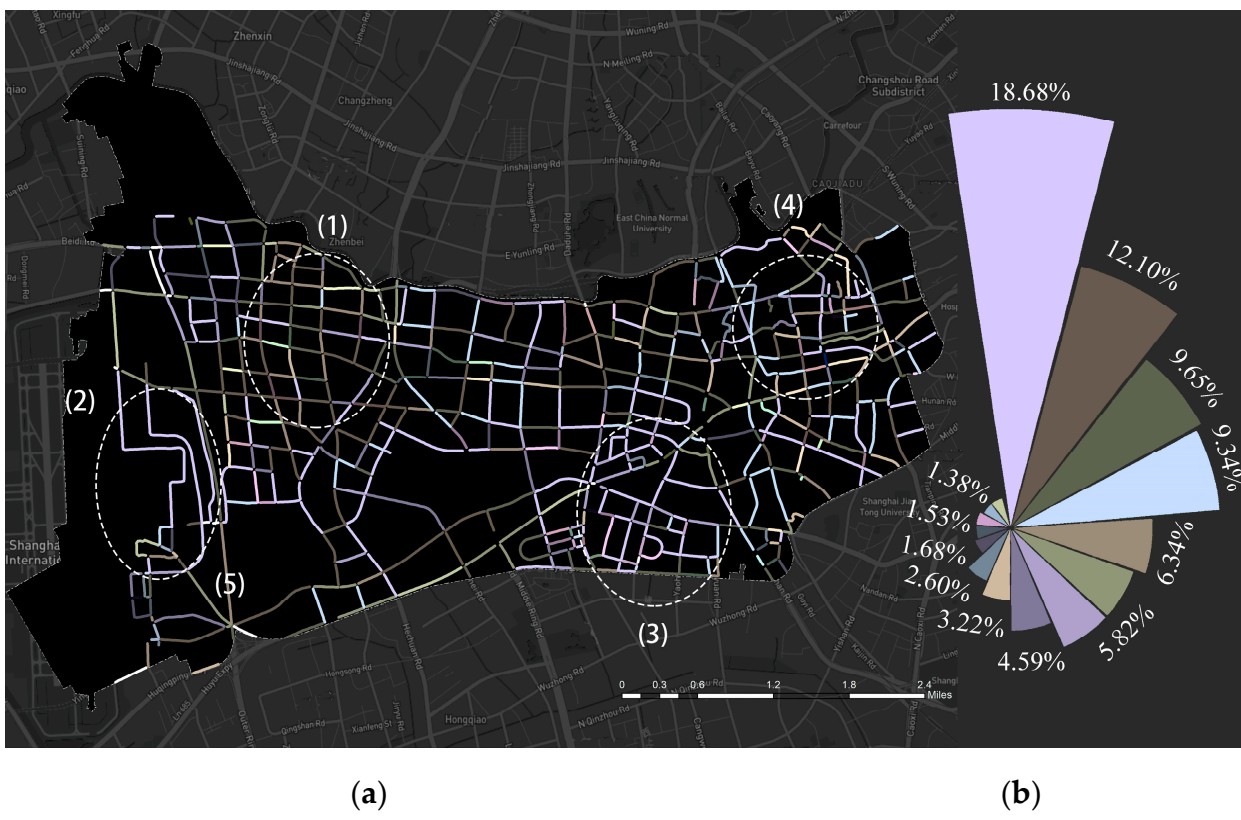

(**a**)                    (**b**)

**Figure 10.** Spatial distribution of street color in Changning District. (**a**) Colors' spatial distribution; (**b**) Top 15 dominant colors.

#### 4.2.2. Building Façade Dominant Color Distributions among Different Functional Streets

We also explored the relationship between three color attributes and street functions. Industrial streets have a lower red hue. Science education streets also have less red–yellow (YR) than other functional streets. The proportion of red–yellow (YR) in greenspace streets is the most compared to other functional areas. The saturation of industrial streets is mainly low, while it is mostly medium–low (ML) in other functional areas. The values of commercial areas are high (H) and medium (M), with the least colors in the medium–high (MH) category. The values of other functional areas are mostly high (H), as seen in Figure 11.

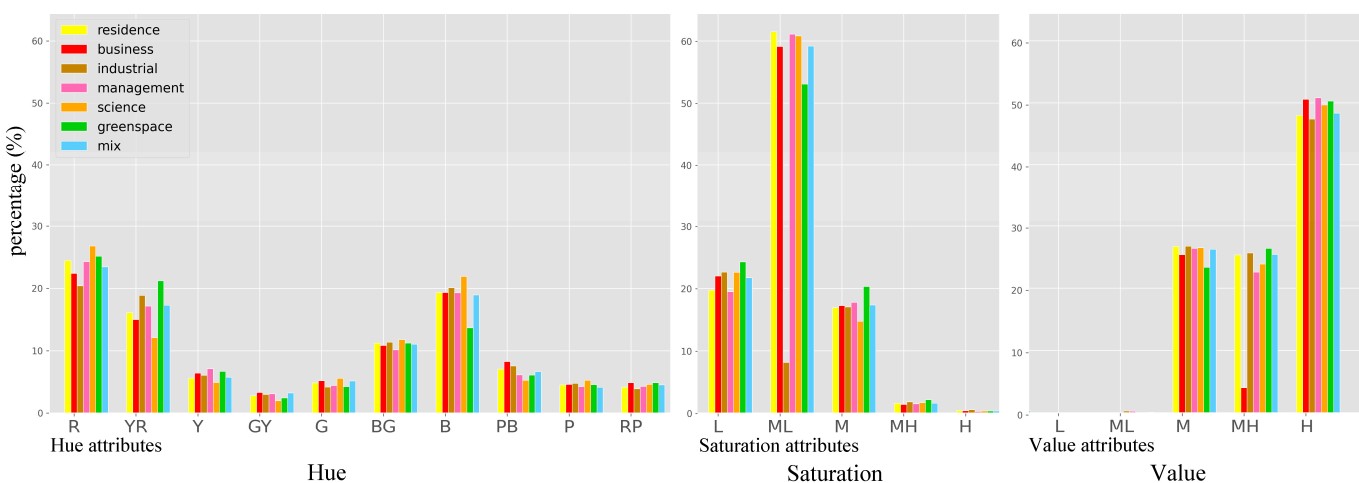

**Figure 11.** Numerical distribution of three channels in different functional streets.

Specifically, in residential streets (a), the dominant colors are light grey blue (H7S4V5), dark grey brown (H1S4V2) and dark grey green (H3S4V2). The color with the highest brightness is light grey yellow (H1S4V5). In business streets (b), medium grey yellow (H1S4V3, H1S4V4) is more prominent. In industrial streets (c), cold colors such as medium grey purple (H7S4V4) and dark grey purple (H7S4V3) are the most frequent. In management streets (d), medium grey yellow (H1S4V4) and light grey purple (H8S4V5) are more prominent. In science education streets (e), medium grey red–purple (H10S4V4, H10S4V3) colors are more prominent. In greenspace areas (f), light grey yellow (H1S4V5) is more prominent than in other types of streets. In mixed functional streets (g), the dark grey red (H1S4V2) and dark grey yellow (H2S4V2) gradually decrease, as shown in Figure 12.

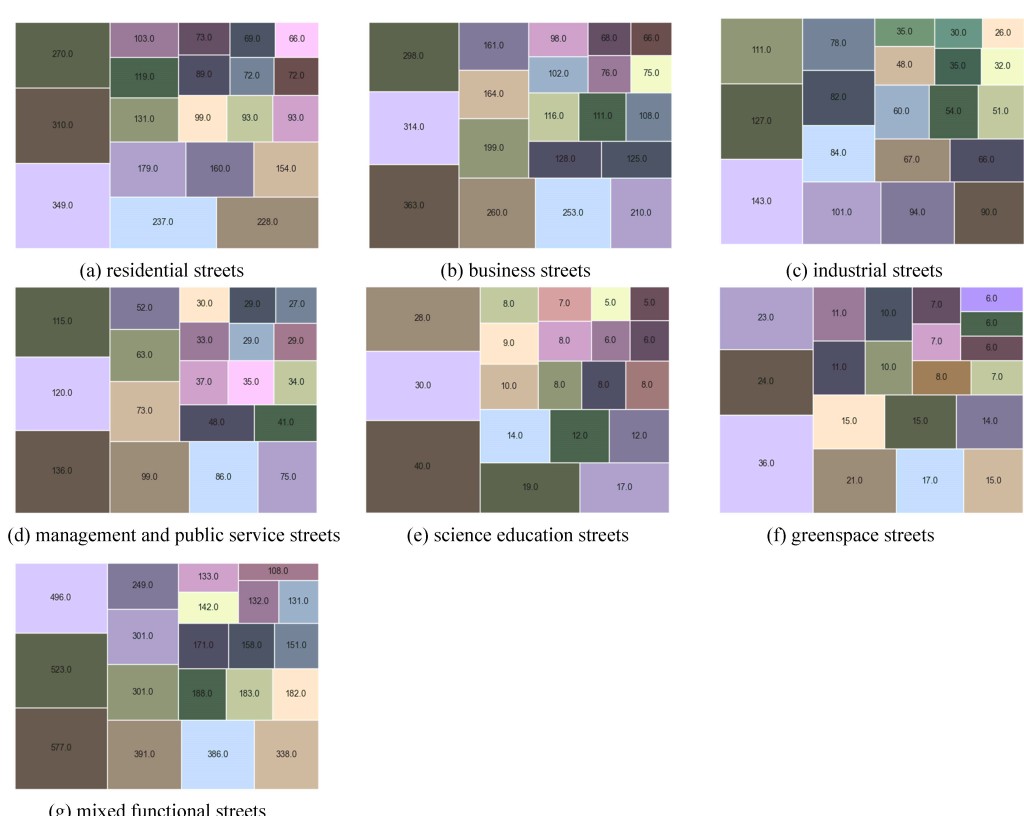

**Figure 12.** Building façade colors in different functional streets.

### 4.3. Color Attributes, Color Harmony, Color Diversity and Street Functions

#### 4.3.1. Logistic Regression Analyses of Color Attributes (HSV) and Street Function

The multinomial logistic regression analysis indicated that in terms of hue, compared with industrial areas, red (R) is more common in residential areas (B = 0.346, $p < 0.01$), science education areas (B = 0.720, $p = 0.001$) and mixed functional areas (B = 0.226, $p = 0.008$). Green (B = 0.473, $p = 0.002$), red (B = 0.324, $p < 0.01$) and red–purple (B = 0.442, $p = 0.005$) are more common in business streets. Blue (B = −0.585, $p = 0.006$) is less common in green areas. In terms of saturation, low saturation is more common in residential streets (B = −0.238, $p = 0.002$) than industrial streets. Management streets' saturation (B = −0.288, $p = 0.002$) is lower than that of industrial streets. In terms of value, compared with industrial streets, management streets (B = 0.293, $p = 0.001$) are more likely to have lower values, as shown in Table 1.

**Table 1.** Multinomial logistic regression analysis.

| Function | | B | SE | Wald | df | Sig. | Exp(B) | 95% Conf. Interval | |
|---|---|---|---|---|---|---|---|---|---|
| | | | | | | | | Lower | Upper |
| Residential | Intercept | 0.662 *** | 0.087 | 57.844 | 1 | 0.000 | | | |
| | [H = R] | 0.346 *** | 0.092 | 14.034 | 1 | 0.000 | 1.413 | 1.179 | 1.694 |
| | [S = L] | −0.238 ** | 0.077 | 9.669 | 1 | 0.002 | 0.788 | 0.678 | 0.916 |
| Business | Intercept | 0.596 *** | 0.087 | 47.055 | 1 | 0.000 | | | |
| | [H = G] | 0.473 ** | 0.155 | 9.283 | 1 | 0.002 | 1.605 | 1.184 | 2.176 |
| | [H = R] | 0.324 *** | 0.092 | 12.369 | 1 | 0.000 | 1.383 | 1.154 | 1.658 |
| | [H = RP] | 0.442 ** | 0.157 | 7.964 | 1 | 0.005 | 1.556 | 1.145 | 2.115 |
| Management | Intercept | −0.314 ** | 0.107 | 8.593 | 1 | 0.003 | | | |
| | [S = L] | −0.288 ** | 0.094 | 9.302 | 1 | 0.002 | 0.750 | 0.623 | 0.902 |
| | [V = L] | 0.293 ** | 0.091 | 10.278 | 1 | 0.001 | 1.340 | 1.121 | 1.603 |
| Science | Intercept | −2.192 *** | 0.210 | 108.496 | 1 | 0.000 | | | |
| | [H = R] | 0.720 ** | 0.213 | 11.493 | 1 | 0.001 | 2.055 | 1.355 | 3.117 |
| Greenspace | Intercept | −1.597 *** | 0.173 | 84.860 | 1 | 0.000 | | | |
| | [H = B] | −0.585 ** | 0.213 | 7.567 | 1 | 0.006 | 0.557 | 0.367 | 0.845 |
| Mixed | Intercept | 1.288 *** | 0.080 | 258.521 | 1 | 0.000 | | | |
| | [H = R] | 0.226 * | 0.086 | 7.004 | 1 | 0.008 | 1.254 | 1.060 | 1.483 |

Note. * $p < 0.01$. ** $p < 0.005$. *** $p < 0.001$. Industrial street is used as a reference. Only significant differences are included.

### 4.3.2. One-Way ANOVA Analyses of Color Harmony, Diversity and Street Function

We conducted two steps to measure color harmony. We first calculated the harmonies between the most and second-most dominant colors (h1), the second- and the third-most dominant colors (h2) and the most and the third-most dominant colors (h3). Second, the mean of h1, h2 and h3 was calculated to represent the color harmony value. Color diversity was computed following a fractal dimension study, as indicated in the Materials and Methods section [35]. We found that the scores of harmony and diversity showed an approximately normal distribution. The average mean value of harmony is 1.68, and the interval range is between −1.14 and 4.12; the first and second colors' harmony interval range is between −1.72 and 4.43; the first and third colors' harmony interval range is between −2.32 and 4.60; and the second and third colors' harmony interval range is between −2.00 and 4.61. The mean value of diversity is 1.84, and the interval range is between 0.98 and 2.35. Regarding color harmony and color diversity, we present the values determined for different degrees in Table 2. The results suggest that the harmony and diversity of 1.5 in all functional areas accounted for the largest proportion, amounting to 26.23% and 70.86%, respectively, as shown in Figure 13a.

**Table 2.** Different degrees of color harmony and color diversity.

| Measure | Scale | Value Range |
|---|---|---|
| Harmony | very harmonious | 3.5, 4 |
| | moderately harmonious | 2.5, 3 |
| | quite harmonious | 1.5, 2 |
| | slightly perceptibly harmonious | 0.5, 1 |
| | moderation | 0, 0.5 |
| | slightly perceptibly disharmonious | −0.5, −1 |
| | quite disharmonious | −1.5 |
| Diversity | slightly perceptibly diversiform | 0.5 |
| | quite diversiform | 1 |
| | moderately diversiform | 1.5 |
| | very diversiform | 2 |

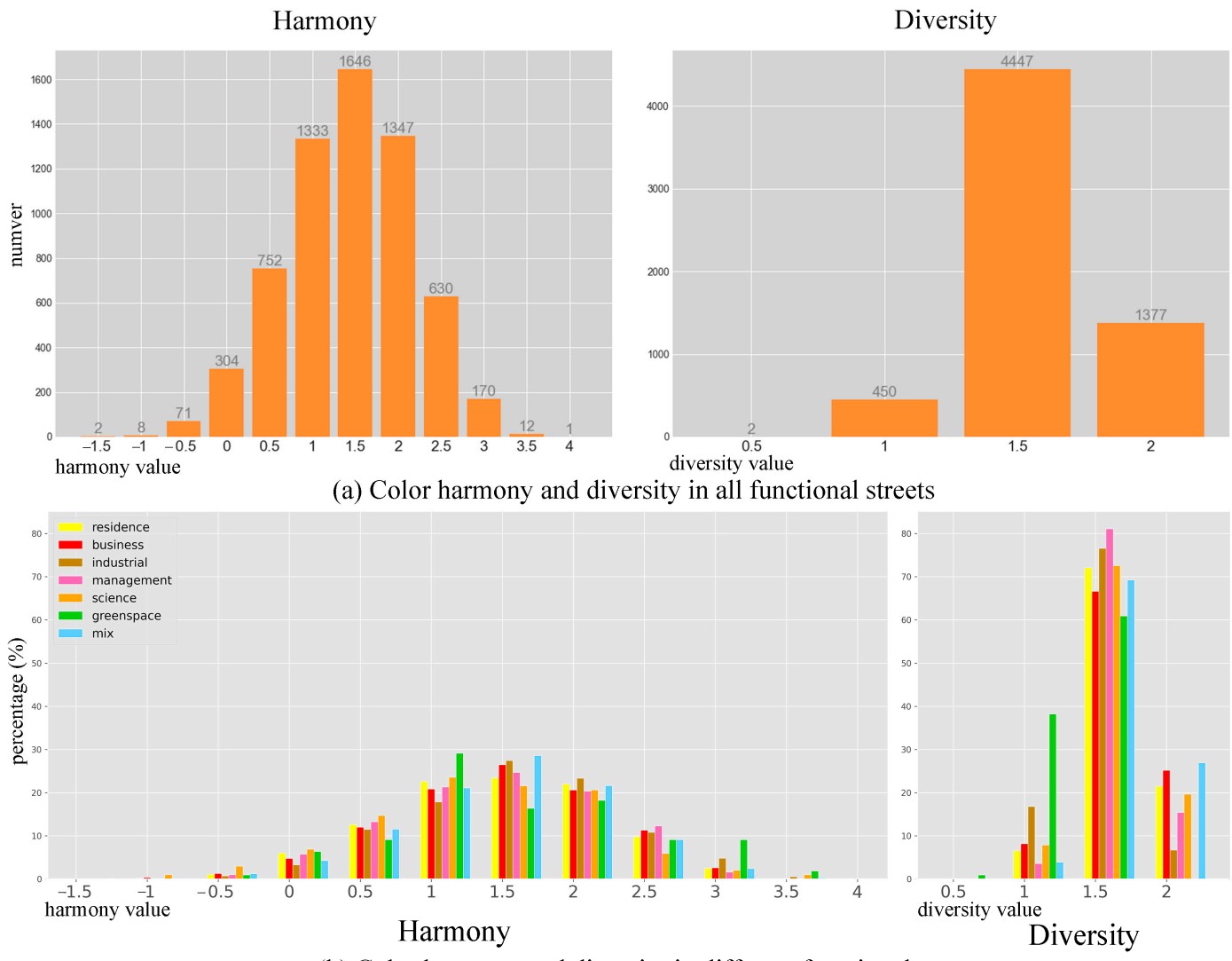

(a) Color harmony and diversity in all functional streets

(b) Color harmony and diversity in different functional streets

**Figure 13.** Color harmony and diversity in (**a**) all functional streets; (**b**) different functional streets.

Measuring the color harmony and diversity in different functional streets, as shown in Figure 13b, we found that the harmony degree was concentrated within the range of slightly perceptibly harmonious (number 1) and quite harmonious (number 2). Residential streets (23.41%), business streets (26.42%), industrial streets (27.40%), management streets (24.65%) and mixed functional streets (28.58%) were quite harmonious (number 1.5), accounting for the largest proportions in each type of street. Science education streets (23.53%) and greenspace streets (29.09%) were mostly found to be slightly perceptibly harmonious (number 1). Regarding the quite harmonious streets (number 2), industrial streets (23.29%) accounted for the largest proportion. Regarding the moderately harmonious streets (number 2.5), management streets (12.23%) accounted for the largest proportion. Regarding the moderately harmonious (number 3) and very harmonious (number 3.5) streets, the greenspace streets constituted the largest proportion, accounting for 9.09% and 1.82%. The diversity of greenspace is more within the scale of quite diversiform (number 1) compared with other functional streets, indicating that the building façade's color diversity in greenspace areas is lower than in others. The mixed functional streets are considered very diversiform, indicating that the colors of the mixed functional streets are very diverse compared with other functional streets. Management streets are mostly found to

be moderately diversiform (number 1.5), indicating that the color diversity is moderate and concentrated.

The homogeneity of variance test indicated that the color diversity and harmony were heterogenous; thus, we applied ANOVA analysis. The results indicated that the average harmony of the three most dominant colors, as well as the harmony between the first and second colors, was more strongly related to the street function ($p < 0.01$).

Using the ANOVA LSD method to compare color diversity and harmony across different functional streets, we found significant differences in diversity and harmony, with $p$-values less than 0.05, as shown in Table 3. For color harmony, we found that industrial streets were more harmonious than residential (MD = 0.1107991, $p = 0.002$), business (MD = 0.1461942, $p < 0.01$), management (MD = 0.1434312, $p = 0.001$), science (MD = 0.2851327, $p < 0.01$) and mixed functional (MD = 0.1137207, $p < 0.01$) streets. There were no significant differences between industrial streets and greenspaces. Regarding color diversity, mixed functional streets were more diversified than residential (MD = −0.0254289, $p < 0.01$), business (MD = −0.0321951, $p < 0.01$) and management (MD = −0.0438900, $p < 0.01$) streets. Industrial streets were less diversified than residential (MD = 0.1343846, $p < 0.01$), management (MD = 0.1159235, $p < 0.01$) and science (MD = 0.1279672, $p < 0.01$) streets. Greenspace streets were less diversified than business (MD = 0.2775607, $p < 0.01$), management (MD = 0.2658658, $p < 0.01$) and science (MD = 0.2779095, $p < 0.01$) streets.

**Table 3.** ANOVA analysis of color harmony and diversity.

| Dependent Variable | (I) Function | (J) Function | Mean Deviation (I–J) | Std. Err. | Sig. | 95% Conf. Interval | |
|---|---|---|---|---|---|---|---|
| | | | | | | Lower | Upper |
| Harmony | Industrial | Residential | 0.1107991 ** | 0.036433 | 0.002 | 0.039378 | 0.1822203 |
| | | Business | 0.1461942 *** | 0.0359673 | 0.000 | 0.0756858 | 0.2167025 |
| | | Management | 0.1434312 *** | 0.0443257 | 0.001 | 0.0565377 | 0.2303248 |
| | | Science | 0.2851327 *** | 0.0783627 | 0.000 | 0.131515 | 0.4387503 |
| | | Mixed | 0.1137207 *** | 0.0338586 | 0.001 | 0.0473462 | 0.1800952 |
| Diversity | Residential | Industrial | 0.1343846 *** | 0.0099198 | 0.000 | 0.1149384 | 0.1538308 |
| | | Greenspace | 0.2843269 *** | 0.0197503 | 0.000 | 0.2456095 | 0.3230443 |
| | | Mixed | −0.0254289 *** | 0.0069292 | 0.000 | −0.0390126 | −0.0118452 |
| | Business | Industrial | 0.1276183 *** | 0.009793 | 0.000 | 0.1084207 | 0.14681605 |
| | | Greenspace | 0.2775607 *** | 0.019687 | 0.000 | 0.2389675 | 0.3161539 |
| | | Mixed | −0.0321951 *** | 0.0067465 | 0.000 | −0.0454205 | −0.0189697 |
| | Management | Industrial | 0.1159235 *** | 0.0120688 | 0.000 | 0.0922645 | 0.1395825 |
| | | Greenspace | 0.2658658 *** | 0.0209124 | 0.000 | 0.2248703 | 0.3068613 |
| | | Mixed | −0.0438900 *** | 0.0097605 | 0.000 | −0.0630239 | −0.024756 |
| | Science | Industrial | 0.1279672 *** | 0.0213362 | 0.000 | 0.0861409 | 0.1697935 |
| | | Greenspace | 0.2779095 *** | 0.0273296 | 0.000 | 0.2243341 | 0.3314849 |

Note. ** $p < 0.005$. *** $p < 0.001$. Only significant differences are included.

## 5. Discussion

### 5.1. Urban Color Distribution in Different Cities

The climate, geography, citizens' preferences and the historical context may play roles in influencing urban color distributions. The dominant urban façade colors identified in our study are consistent with some of the previous studies. For instance, a study compared three metropolises and found that the dominant urban color was more likely to be blue in Shanghai [19]. Shenzhen City's urban colors are grey tones with low chroma [16], with similar chroma to the dominant colors identified in the present study. In Wuhan City, the

dominant urban colors are mainly yellow–red and purple [51], which is in line with our study. Nguyen and Teller found that in 18 areas of Lie'ge City, there were four types of colors, namely light beige, grey orange, grey red and light grey [17], which is similar to our study.

Our analysis results are inconsistent with some studies, which might be related to the location skewness in building colors. The climate, geography, citizens' preferences and historical context may play roles in influencing urban color distributions. For example, Busan City's color guidelines suggested 12 main urban colors, including green, which is different from the dominant colors that we identified in this study [52]. This difference might because Busan City is surrounded by valleys and rivers, and the green color is more harmonious with the natural surroundings. A study conducted in Beiwaitan, Shanghai, found that grey green and red colors are among the most popular colors in this area, and the architecture has undergone five chronological phases dominated by distinctive colors. For instance, between 1920 and 1937, decorative-style buildings were popular in this area, with the main colors of grey green [22]. However, historical buildings in our study area were mostly garden houses with yellow–red colors in Changning District [29]. These results suggest that the changes in architectural design styles across history have an important influence on urban color distributions.

### 5.2. Street Function and Building Façade Color

According to the above results, we found that the building façade color distribution is related to the street function, which might be related to the color associations of citizens. Namely, when seeing certain colors, we may think of some items. Blue is the dominant color in residential streets in Changning District. Such a result is consistent with previous research conducted in the USA, in which it was found that when decorating their residences, many people choose blue or a less saturated color as the main color [7]. Thus, color associations are related to the street function, and people may choose certain colors unconsciously, according to the objects that are associated with these colors.

Building façades' color harmony and diversity can significantly influence perceptions and preferences of the urban environment. In this study, we found the urban street functions are related to the harmony and diversity of urban colors. We found the industrial streets had less red but were more harmonious than science education areas, residential areas and mixed functional areas. Such a result highlights the importance of addressing the three color attributes when considering color harmony. However, only a few studies have addressed street functions in examining urban color harmony and diversity. Li, Yang et al. found that if a color tends toward green or blue, the color harmony is lower [14]. This might be why we found that colors in greenspace streets are more harmonious, with a greater proportion classified as very harmonious. Zhou, Zhong et al. argued that the building colors of historic districts in the core area were harmonious with the regional building façade colors in Guangzhou because of the preservation policies [53].

### 5.3. Strengths and Limitations

A strengths of this study is that we addressed street functions when discussing the urban color distribution, color harmony and color diversity. We used advanced technologies, such as street view image crawling, image semantic segmentation and the K-means algorithm, to examine street building façades' color distribution. We also explored the color harmony and color diversity among building façades' dominant colors in relation to street functions. This article provides a new direction for urban color management, adding dominant color, color harmony and diversity indexes from the perspective of urban functional streets. This study has three main limitations, which should be addressed in future explorations. First, advanced neural network technology, such as Transformers, could be applied to more accurately identify windows and doorways on the building façades. Second, we only excluded the colors with values less than 50 in all three RGB channels, and a more effective method is needed to diminish the possible influence of

shadows in building façades. Third, a more accurate land-mix index, such as information entropy, should be applied to classify street functions more precisely.

## 6. Conclusions and Implications

This study utilized street view images to explore the urban color distribution across different functional streets. We found that the dominant urban colors in Changning District are blue and red–yellow, with medium–low saturation and low brightness. Considering street functions, we found that in terms of hue, red is relatively rare in industrial streets; green (B = 0.473, $p$ = 0.002), red (B = 0.324, $p$ < 0.01) and red–purple (B = 0.442, $p$ = 0.005) are more common in business streets; red–yellow is less prominent in science streets than in other functional streets, and blue (B = −0.585, $p$ = 0.006) is relatively less prevalent in greenspaces. As for saturation, that of residential streets (B = −0.238, $p$ = 0.002) is higher than that of industrial streets. Management streets (B = −0.288, $p$ = 0.002) are lower than industrial streets in terms of saturation. The values of different functional streets are mostly high, and management streets (B = 0.293, $p$ = 0.001) have lower values than industrial areas. Compared with the three attributes of color itself, we found that the harmony and diversity are more related to the street function. Industrial streets are more harmonious than other functional streets, except for greenspaces. Residential (MD = 0.1343846, $p$ < 0.01), business (MD = 0.2775607, $p$ < 0.01) and management (MD = 0.0438900, $p$ < 0.01) streets are more diversified than industrial, greenspace and mixed functional streets. Science streets are more diversified than industrial (MD = 0.1279672, $p$ < 0.01) and greenspace streets (MD = 0.2779095, $p$ < 0.01), and there are no significant differences from mixed functional streets.

This study highlights the significant relationship between street functions and color attributes, as well as measuring the color distribution, harmony and diversity with advanced techniques. Such a result implies that we should develop urban color planning strategies that are integrated with the street functions associated with the buildings. Moreover, the relationships among different colors should be considered in urban color planning, such as color harmony and color diversity, rather than only focusing on a single color.

**Author Contributions:** Conceptualization, Yujia Zhai and Ruoyu Gong; methodology, Ruoyu Gong, Junzi Huo and Binbin Fan; data curation, Ruoyu Gong; writing—original draft preparation, Yujia Zhai and Ruoyu Gong; writing—review & editing, Yujia Zhai and Ruoyu Gong; visualization, Ruoyu Gong and Junzi Huo; project administration, Yujia Zhai; funding acquisition, Yujia Zhai. All authors have read and agreed to the published version of the manuscript.

**Funding:** This research was funded by the Fundamental Research Funds for the Central Universities, grant number 22120220302 and Shanghai Science and Technology Innovation Project by 22692111700.

**Data Availability Statement:** Not applicable.

**Acknowledgments:** We would like to thank the editors and the anonymous reviewers for their constructive suggestions and comments, which helped to improve this paper's quality.

**Conflicts of Interest:** The authors declare no conflict of interest.

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
