# Peer review of "Building Façade Color Distribution, Color Harmony and Diversity in Relation to Street Functions: Using Street View Images and Deep Learning"

_ijgi, doi:10.3390/ijgi12060224_

Round 1

Reviewer 1 Report

Review Report

Building facade color distribution, color harmony and diversity in relation to street functions: Using street view images and deep learning

This paper investigates the relationships between building façade color and street functions from the viewpoint of the street. It proposes a comprehensive research method flow that will be helpful for urban street planning.

I recommend this paper can be published with some revisions.

1. Building façade color is highly susceptible to shadows. In Figures 5(b) and 6, the color looks darker than expected in well-lit conditions. However, saturation and value, especially value, may be more affected by light than the building façade color itself. As a result, how to cope with the uneven light distribution should be considered.

2. Page 2, Line 61 to 66: The authors stated that the second main contribution is using new technologies. However, the three techniques mentioned here are all existing and have been applied in street view research.

3. Figure 1: Please revise to distinguish between the method and data in the flowchart. In addition, please check the letter case.

4. Figure 4: The legend of the segment street view images lacks a few colors, such as magenta. Please check.

5. Figure 9: It is necessary to interpret what the x-axis and y-axis indicate. (a)(b)(c) should all be annotated uniformly. Moreover, why are (a) and (b) chosen to be presented in different forms? If one emphasizes V and one emphasizes S, the same form of the chart should be used.

6. Page 7, Line 230 to 235: The formulas and methods for measuring the urban color characteristics should be displayed here.

7. Page 14, Line 396 to 399: The authors concluded that the results of this study demonstrate blue and the least saturated color as the dominant color in residential areas. However, this seems less convincing in Figure 12 and Table 1. Please check.

8. Some minor issues. Please check and unify the façade or facade. Please place the citations alongside the formulas. Page 4, Line 145 to 151: km2 should be km2. Figure 7: Please change the background color and layout of the legend to make it clearer. Figure 11 & Figure 13(b): Using the same y-axis is strange because there is no comparability between Hue and Saturation and Value, nor between Harmony and Diversity, so they should not be laid out side by side.

Author Response

We sincerely thank the reviewer’s valuable comments. We have revised our main contributions, corrected mistakes in Figure 1 and 4, and added the meaning of x-axis and y-axis in Figure 9. Formulas and methods of color harmony and diversity are been further explained in 3.3.3 part. Our results are been checked to make it seems more convincing and revised some minor issues.

Please see more details in the attachment.

Reviewer 2 Report

1. The literature review part is not very targeted, for example, Existing Urban Color Planning in China, so it is suggested to revise the literature review comprehensively, so as to better support your study questions.

2. It is suggested to adjust the structure of the study. The third part should be materials and methods, and Figure 1 should be the research process. The author needs a more detailed explanation of the study process, rather than the present form.

3. The author has too much content in the method part. In fact, the method part should be the experimental process, not the specific method explanation. Therefore, it is suggested that most of the content here can be adjusted to the first part of the result, and then the method part should be rewritten.

4. Just as the author said, although the author wrote a lot in the result part, I don't think the current result part can get your conclusion, in other words, your conclusion is not targeted.

5. There is too little content in the discussion section, and the comparison with other studies, contributions and limitations of this study have not been elaborated in detail.

Author Response

Thanks for the valuable suggestions. Following the reviewer’s comment, we have reframed the structure of the literature review and discussions parts. We also added many literature reviews and elaborated the contributions and limitations in detail. The third part is been adjusted from methodology to materials and methods, and we removed some excessive descriptions in methods part. Our study process is explained in more details.

Please see more details in the attachment.

Reviewer 3 Report

The subject assessed in the article is of interest for the urban and landscape design field, as well as the urban planning policy makers. The authors have worked on using recent digital technologies to potentiate the ability of a surveyor to gather information from a large are on the building’s façade coloring.  The authors remark the importance of such work in relation with use of the space and functions, attractiveness, and harmony. They have presented the results for a specific area, that has not been studied, to my and to the authors’ knowledge, in such a granular yet vast scale. Presenting conclusive results on the color predominance with tested statistical significancy for different street functions.

However, authors should improve the actual form of the manuscript before it is published, to arrive to its full potential and to make it more understandable to the reader. In general, the author might be citing too often on things that are not required, there are some phrases in which the English could be improved, the impact of the work is not highlighted fully and some results are presented within the methodology. Finally, given the subject of the special issue, more references to the geographical conditions and settings should be exposed.

More in specific, the state of the art described in the introduction is rather complete, however some additions regarding the impact that color can have on the population could be presented (e.g. color-emotions, color-temperature perception); the methodology has some passages which are hard to follow, and some partial results have been included within; and, the discussion section could be extended to better comment the obtained results (e.g., context and location skewness). Finally, in the conclusion a more direct relation with the relevance of the obtained result could be done, highlighting the objective and quantitative description of a city’s area.

I believe the authors, should also consider the following specific comments if they wish to improve their work:

1. At the beginning of the introduction (liens 27-29) the importance of color in a city is mentioned without a particular way of measuring or relating it objectively and quantitatively. Without such criteria, the impact of the work losses a bit of weight.

2. What type of effects generate the less hue, less saturation and high brightness described in lines 51-53?. Please elaborate in the text.

3. In line 140, better specify “… main parts, as described in Figure 1.”

4. In line 142 the acronym POI is used, without a previous definition.

5. In lines 185-188, Is it necessary to cite all this references? Please revise the whole document.

6. In lines 207-208, revise English the phrase is not so clear.

7. In line 223, you mention the use of distance between colors, which distance are you calculating giving that the color models are not linear?

8. The process described within lines 242-256 is complex, perhaps presenting one example could help the reader?

9. Isn’t Figure 7 a partial or preliminary result? Should it be within the methodology?

10. In lines 396-398, you mention the link between the results and the location, please elaborate more in such relationship as it is not meant to be a worldwide phenomenon. Are there any diverse results in literature?

11. Revise lines 402-406, there is a repeated phrase.

12. Lines 405-406 seem to contradict, “… colors in green space streets are more harmononious…” the previous phrase had just mention that green colors are less harmonious. How to explain this?

Author Response

We sincerely thank the reviewer for the kind reminder. We have highlighted our works in introduction and conclusion, added more geographical conditions and settings in our study area. Some examples are presented in part 3.4 to explain our methods clearly. We have elaborated more in discussions about color distribution differences in worldwide. Finally, we have checked that all references are relevant to the contents of the manuscript and revised our English.

Please see more details in the attachment.

Round 2

Reviewer 2 Report

Accept in present form